Identification and immunoinfiltration analysis of key genes in ulcerative colitis using WGCNA

Ni Siyi
Liu Yingchao
Zhong Jihong
Shen Yan 20114022@zcmu.edu.cn
Department of Gastroenterology, The Second Affiliated Hospital of Zhejiang Chinese Medical University , Hangzhou , China
Uversky Vladimir
Electronic publication date: 2024 Feb 26
Publication date: 2024
Volume: 12
Electronic Location ID: e16921
Received 2023 Nov 21; Accepted 2024 Jan 19
Copyright: ©2024 Ni et al.
Copyright year: 2024
Copyright holder: Ni et al.
License: This is an open access article distributed under the terms of the Creative Commons Attribution License, which permits unrestricted use, distribution, reproduction and adaptation in any medium and for any purpose provided that it is properly attributed. For attribution, the original author(s), title, publication source (PeerJ) and either DOI or URL of the article must be cited.
License URL: https://creativecommons.org/licenses/by/4.0/

Keywords: Ulcerative colitis, Bioinformatics analysis, WGCNA, Immunoinfiltration analysis

Funding: Zhejiang Medical Association Project 2019ZYC-B01 Zhejiang Traditional Chinese Medicine Science and Technology Program 2023ZL061 This study was supported by the Zhejiang Medical Association Project (2019ZYC-B01) and the Zhejiang Traditional Chinese Medicine Science and Technology Program (2023ZL061). The funders had no role in study design, data collection and analysis, decision to publish, or preparation of the manuscript.

==============================
Objective

Ulcerative colitis (UC) is a chronic non-specific inflammatory bowel disease characterized by an unclear pathogenesis. This study aims to screen out key genes related to UC pathogenesis.

Methods

Bioinformatics analysis was conducted for screening key genes linked to UC pathogenesis, and the expression of the screened key genes was verified by establishing a UC mouse model.

Results

Through bioinformatics analysis, five key genes were obtained. Subsequent infiltration analysis revealed seven significantly different immune cell types between the UC and general samples. Additionally, animal experiment results illustrated markedly decreased body weight, visible colonic shortening and damage, along with a significant increase in the DAI score of the DSS-induced mice in the UC group in comparison with the NC group. In addition, H&E staining results demonstrated histological changes including marked inflammatory cell infiltration, loss of crypts, and epithelial destruction in the colon mucosa epithelium. qRT-PCR analysis indicated a down-regulation of ABCG2 and an up-regulation of IL1RN, REG4, SERPINB5 and TRIM29 in the UC mouse model. Notably, this observed trend showed a significant dependence on the concentration of DSS, with the mouse model of UC induced by 7% DSS demonstrating a more severe disease state compared to that induced by 5% DSS.

Conclusion

ABCG2, IL1RN, REG4, SERPINB5 and TRIM29 were screened out as key genes related to UC by bioinformatics analysis. The expression of ABCG2 was down-regulated, and that of IL1RN, REG4, SERPINB5 and TRIM29 were up-regulated in UC mice as revealed by animal experiments.

Introduction

Ulcerative colitis (UC) is a chronic non-specific intestinal disease  (Wils & Peyrin-Biroulet, 2023) clinically characterized by symptoms such as diarrhoea, hematochezia, stomach ache and vomiting. At present, the exact etiopathogenesis of UC remains unclear (Tan et al., 2020; Adams, Close & Shreenath, 2022). As is known, the morbidity of UC is relatively higher in Europe, and is on the upswing in south (Haberman et al., 2019; Fries et al., 2022). UC progression attributes to a multifaceted complex interplay of factors, encompassing genetics as well as environmental elements including lifestyle, diets, hygiene practices and intestinal microbiome (Liu et al., 2022; Li et al., 2022a; Zhang et al., 2020). Experimental studies have demonstrated massive infiltration of neutrophil granulocytes, NK cells, IL-2, Th17 cells, TNF- α, IFN- γ, IL-17A, IL-22, etc. in the intestinal mucosa of UC patients (Ardizzone et al., 2009; Zhang et al., 2022; Lai, Shen & Ran, 2019).

The occurrence of delayed diagnoses in inflammatory bowel disease is common and correlates with unfavorable outcomes. Early detection and timely intervention are key to improving disease outcomes and maximally optimizing prognosis (Agrawal et al., 2021). Encouragingly, bioinformatics analysis makes it possible to explain the molecular mechanisms of multiple diseases. It facilitates the identification of novel biomarkers, thus enhancing the diagnostic and prognostic approaches for UC (Cheng et al., 2020; Dong et al., 2022; Yang et al., 2022). Weighted gene co-expression network analysis (WGCNA) is conducive to the analysis of multiple gene expression patterns, and its clustering criteria hold biological significance (Huang et al., 2022; Li et al., 2022b). The unique soft-threshold algorithm of WGCNA guides the gene expression network towards a scale-free network distribution, resulting in highly reliable outcomes. It enables the clustering of genes based on similar gene expression patterns to form modules, and facilitates analysis of the relationship between modules and specific characteristics (Zhou et al., 2022; Su et al., 2022).

In this research, relevant differentially expressed genes (DEGs) were confirmed, followed by construction of weighted co-expression networks using WGCNA, with an aim to screen out modules and gene sets associated with corresponding clinical features. Enrichment analyses of the key genes were further carried out using GO and KEGG. Immune cell infiltration was assessed using CIBESORT. To test the reliability of bioinformatics analysis, a UC mouse model was constructed, and the expression of key genes was detected. Our findings are expected to mark a new direction for finding fresh therapeutic targets for UC.

Materials and methods

Data source

The “GEOquery” package in R software was used to download relevant gene expression datasets from the Gene Expression Omnibus (GEO) database. The downloaded datasets include GSE38713 (comprising 15 UC active samples, 15 UC remission samples, and 13 normal control samples), GSE42911 (consisting of five UC samples and five normal control samples), GSE134025 (comprising three UC samples and three general control samples), and GSE87466 (comprising 87 UC samples and 21 general control samples).

Data preprocessing and DEG screening (Hu et al., 2022)

The three downloaded gene expression datasets, GSE38713, GSE42911 and GSE134025, were combined, after which batch effect removal was conducted using the “sva” package to eliminate differences between batches. The effectiveness of batch effect removal was visualized using the Quartile-Quantile plot (Q–Q plot) (Fig. S1), and a two-dimensional PCA clustering plot was generated to demonstrate the normalization between batches (Fig. S2). DEGs were screened using the “limma” package. For visualization of the differential expression of the DEGs, the “ggplot2” package was used to generate volcano plots, and the “pheatmap” package was used for plotting heatmaps. P < 0.05 and |log2FC| > 2 were regarded to be statistically significant.

WGCNA

The gene expression matrix was sorted in descending order based on the variance, and the top 5000 genes having the highest variance were analyzed via WGCNA using the “WGCNA” package in R, with an aim to discern modules and gene sets related to clinical features. Abnormal samples were excluded to enhance the reliability of network construction. We set a soft-thresholding power with a scale-free R2 close to 0.9 and a slope close to 1 for transforming the adjacency matrix into a topological overlap matrix, and adjusted the soft threshold power to 14 and the minimum module size to 30 for constructing networks and detecting modules.

Functional enrichment analysis (Kanehisa, 2019; Kanehisa et al., 2023)

The “clusterProfiler” in R package was used for performing GO and KEGG enrichment analysis. GO function enrichment analysis revealed the functional characters of host target genes from three hierarchies based on categories of cellular components, biological processes, and molecular functions.

Immunoinfiltration analysis (Yin et al., 2022; Wu et al., 2022)

The “CIBERSORT” in R package was used for conducting immunoinfiltration analysis on the UC and general control samples. Samples with a p-value < 0.05 were selected, and the distribution of 22 kinds of immunocytes in the samples were obtained. Furthermore, a boxplot was plotted to compare the differential distribution of immunocytes between the UC and normal samples.

Key gene analysis

Obtaining key genes through crossover of the DEGs with the genes identified in the key modules of WGCNA based on the criteria of GS > 0.6 & MM > 0.8. Tissue expression boxplots and ROC curves were generated for the key genes. Furthermore, relevance analysis was conducted between the key genes and differential immunocytes, with a heatmap plotted for illustrating the correlation. Eventually, the expression of the key genes was validated in the GSE87466 dataset.

Animal model construction (Wang et al., 2019)

Thirty-eight-week-old C57BL/6 mice (male, 20-25g) were raised in a SPF animal installation. The mice were allocated into three groups through random selection: NC group, 5%-dextran sulfate sodium salt (DSS) group and 7%-DSS group (n = 10). The mice in the modeling group were given autoclaved water containing 5% or 7% DSS, while those in the NC group were only provided with autoclaved water, which was changed every other day. Daily observations were made to record the body weight, stool consistency and color, as well as presence of rectal bleeding of mice for the calculation of disease activity index (DAI). On the 7th day, euthanize the mice, and their colon tissues were collected, with gross pictures taken. Subsequently, the total length of the mouse colon was measured. The distal colon was fixed with 4% formaldehyde, and subjected to conventional paraffin embedding, sectionalization, and H&E staining. The remaining colon tissues stored at −80 °C. The animal experiments in this study were approved by the Zhejiang Experimental Animal Ethics Committee (approval number: ZJCLA-IACUC-20020046).

H&E staining

Perform standard H&E staining on dewaxed colon tissue sections and observe histological changes under a microscope.

Quantitative real-time PCR

Collect RNA from colon tissue and complete reverse transcription. The relative expression levels of the five identified key genes (ABCG2, IL1RN, REG4, SERPINB5 and TRIM29) were detected using the HieffqPCR SYBRGreen Master Mix kit (11201ES08, Yisheng Biology, Shanghai, China). The primer sequences are shown in Table 1.

Table 1 Primer sequence information in the experiment.

Primer name	Sequence	
ABCG2	Forward:CATCACATCACCTATCGAGTGA	
Reverse:CTTTCCTTGCTGCTAAGACATC	
IL1RN	Forward:TGGATCATGGCAGGTGCTTGTTC	
Reverse:AGGGGTAGGGTGGGTGGTAGAG	
REG4	Forward:GAGAAACCTGCCTGTGTGGATTGG	
Reverse:GCTTCACTCTTTGTCCTGGGATTCC	
SERPINB5	Forward:CTCTTTGAAACTTGTCAAGCGA	
Reverse:CCTTTCGTTTCTTCCAGTTTGT	
TRIM29	Forward:GAGAAGCAGAAGGAGGAAGTAC	
Reverse:AGTAATTGCTCATCAATGCACC	

Cell model construction

Caco-2 cells (Human colorectal adenocarcinoma cells) (CL-0050, Procell, Wuhan) were cultured in RPMIS 1640 medium (10% FBS, 1% P/S) under a circumstance with 5% CO2 at 37 °C. Caco-2 cells were treated with lipopolysaccharide (LPS, 1 µg/mL) for 24 h to construct UC cell model in vitro.

Cell transfection

When the fusion rate of Caco-2 cells treated with LPS reaches 60–80%, si-NC/si-IL1RN, si-NC/si-REG4, si-NC/si-SERPINB5, si-NC/si-REG4 (synthesized from the Gemma gene) were transfected into the cells for knock-down of IL1RN, REG4, SERPINB5, REG4 using Lipofectamine 2000 co-transfection reagent. Additionally, construct ABCG2 plasmid and transfected into the cells to induce overexpression of ABCG2.

Western blot

The total protein of colon tissue and Caco-2 cells was extracted, and the same amount of protein was separated by gel electrophoresis of polypropylene. Primary antibodies (ABCG2 (4477S; Cell Signaling Technology, Danvers, MA, USA), IL1RN (PA5-95456, IL1RN), IL1RN (PA5-95456, IL1RN), IL1RN (PA5-95456, IL1RN); Thermo Fisher Scientific, Waltham, MA, USA), REG4 (37270S; Cell Signaling Technology), SERPINB5 (9117S; Cell Signaling Technology), TRIM29 (ab244380, Abcam), Bax (2772T; Cell Signaling Technology), Bcl-2 (3498T; Cell Signaling Technology) and Cleaved Caspase-3 (9661T; Cell Signaling Technology) were incubated overnight at 4 °C.

Detection of cell viability

The CCK-8 kit (CA1210; Solarbio, Beijing, China) was used for the detection of cell viability in each group, detection of absorbance value at 450 nm.

Detection of apoptosis

Detection of apoptotic cells using TUNEL assay kit (T2130, Solarbio). Fix and penetrate each group of cells, then add 150 µL Incubate TUNEL reaction solution for 1 h (37 °C) and observe under a fluorescence microscope.

Statistical analysis

Data conforming to the normal distribution were presented as mean ± standard deviation (SD). The GraphPad 8.0 software was applied for performing data analysis. A comparative analysis of difference was conducted, with t test for between-group comparison, and bidirectional analysis of variance (ANOVA) for multiple group comparison. P < 0.05 was regarded to be statistically remarkable.

Results

Data preprocessing and DEG screening

The three downloaded gene expression datasets were merged into a single file, and batch normalization was then performed using the ComBat function of the “sva” package. Subsequently, this integrated dataset, including 13,071 genes, was used for further analysis. Following data preprocessing, a total of 44 DEGs were extracted from the gene expression array using R software, as shown in the volcano plot (Fig. 1A) and heat map (Fig. 1B), with 34 up-regulated and 10 down-regulated DEGs. These DEGs were subsequently subjected to GO and KEGG functional enrichment analyses, with results shown in Figs. 1C and 1D. GO analysis revealed gene enrichment in pathways including humoral immune response, antibacterial humoral response, receptor ligand activity, and signal receptor activator activity. KEGG analysis showed gene enrichment in the IL-17 signal pathway, the cytokine–cytokine receptor interplay, the chemokine signal pathway and the TNF signal pathway.

Figure 1 Differential gene screening and functional enrichment analysis.

(A) Volcanic maps showing the results of differential analysis; (B) the heat map showing the expression of differential genes in each sample; (C) GO functional enrichment analysis of DEGs; (D) enrichment analysis of KEGG pathways of the DEGs (only the top ten items are shown in the figure).

WGCNA

A soft threshold of 14 (Fig. 2A) was set for the construction of a weighted co-expression network (Fig. 2B). Based on correlation between features (Control and UC) and network modules, a correlation heat map (Fig. 2C) was drawn. The correlation between the blue module or the turquoise module and UC was found to exceed 0.5, with a p value below 0.01. Then, taking UC as the target feature, a bar chart (Fig. 2D) was generated to illustrate the importance of different modules for UC. The black, blue and brown modules exhibited a gene significance (GS) surpassing 0.3. The module having the strongest correlation with UC was defined as the key module, namely, the blue module. Then, we plotted a scatter plot of gene connectivity and UC importance for the blue module (Fig. 2E), cor = 0.5, p < 0.01. Genes within the blue module (1,064 genes) were analyzed via functional enrichment analysis, and corresponding GO and KEGG diagrams were obtained (Figs. 3A, 3B). GO analysis revealed gene enrichment in lipid catabolism, mitochondrial intima and phosphate ester hydrolase activities. KEGG analysis showed gene enrichment in oxidative phosphorylation, peroxisome, fatty acid degradation and citric acid cycle (TCA cycle). A total of 14 genes, namely ABAT, ABCG2, ANK3, CDC14A, EDN1, IL1RN, KDELR3, PDK2, REG4, SERPINB5, SPNS2, TRIM29, USP30 and ZNF575, were identified as key genes within the blue module based on the criteria of GS > 0.6 & MM > 0.8.

Figure 2 Weighted Co-expression Network Analysis (WGCNA).

(A) Soft threshold selection; (B) construction of a weighted co-expression network diagram; (C) heat maps of the correlation between features (Control and UC) and modules; (D) bar chart of the importance of different modules to UC; (E) scatter plot of blue module gene connectivity and UC gene significance.

Figure 3 Functional enrichment analysis of the blue module genes.

(A) GO functional enrichment analysis of the module genes; (B) enrichment analysis of the KEGG pathways of the module genes (the figure only shows the top ten items).

Key gene analysis

After taking intersection of the 14 key module genes with the 44 DEGs, five key genes: ABCG2, IL1RN, REG4, SERPINB5 and TRIM29 were obtained (Figs. 4A–4B). Boxplots (Fig. 4C) and ROC maps (Fig. 4D) were generated depicting the tissue expression of the five key genes. Compared with the control group, ABCG2 was down-adjusted in UC patients, while IL1RN, REG4, SERPINB5 and TRIM29 were up-regulated in UC patients. The ROC curve indicated that the AUCs were all >0.85, indicating that all the five key genes had high diagnostic values.

Expression verification of the key genes

In GSE87466, the expression of the five key genes was tested (Figs. 5A–5E), and the results were completely consistent with the aforementioned findings. ABCG2 was down-regulated in UC patients, while IL1RN, REG4, SERPINB5 and TRIM29 were up-regulated in UC patients.

Analysis of immunocyte infiltration

Infiltration analysis of the UC and general control samples was conducted using the “CIBERSORT” in R. Samples with p value <0.05 were selected, and the distribution of 22 kinds of immunocytes in the samples were obtained (Fig. 6A). Subsequently, a comparative analysis of immunocyte differences between the UC and general control samples were performed using the boxplot (Fig. 6B). Significant differences were observed in seven types of immunocytes between the UC and general control samples. Compared to the general control samples, the UC samples showed less infiltration in regulatory T cells (Tregs), macrophages M2, and eosinophils. Conversely, macrophage M1, activated dendritic cells, activated mast cells, and neutrophils exhibited higher infiltration in the UC samples.

Figure 4 Analysis and screening of key genes.

(A) Venn diagram plotted for screening key genes; (B) protein interaction network diagram; (C) box plots of the expression of the five key genes (ABCG2, IL1RN, REG4, SERPINB5 and TRIM29) in the dataset; (D) ROC curves of the five key genes.

Figure 5 Expression verification of the key genes in GSE87466.

(A) Expression box diagram of ABCG2; (B) expression box diagram of IL1RN; (C) expression box diagram of REG4; (D) expression box diagram of SERPINB5; (E) expression box diagram of TRIM29.

Figure 6 Analysis of immune cell infiltration.

(A) Distribution of 22 kinds of immune cells in the sample; (B) analysis of the difference of immune cells in the UC group and normal control group.

Correlation analysis of the key genes and infiltrating immunocytes

We analyzed the correlations between the five key genes and the seven differential immunocytes, and plotted corresponding correlation diagrams (Figs. 7A–7F). In terms of the relationship between immunocytes, dendritic cell activation was positively correlated with neutrophil activation; neutrophils positively correlated with mast cell activation and macrophage M1, but negatively with eosinophils and macrophage M2; mast cell activation was in positive correlation with macrophage M1, but in negative correlation with macrophage M2; macrophage M1 showed a negative correlation with macrophage M2 and T cell regulation. With regard to the relationship between key genes and differential immunocytes, ABCG2 was negatively associated with dendritic cell activation, mast cell activation, macrophage M1 and neutrophils, but positively associated with T cell regulation; IL1RN and REG4 were positively linked with dendritic cell activation, mast cell activation, macrophage M1 and neutrophils, but negatively linked with macrophage M2, T cell regulation and eosinophils; SERPINB5 and TRIM29 were in positive relation to dendritic cell activation, mast cell activation, macrophage M1 and neutrophils, but in negative relation to macrophage M2 and T cell regulation.

Figure 7 Analysis of the relationship between key genes and differential immune cell regulation.

(A) Heat map of the correlation between key genes and differential immune cells; (B) bar chart analysis of the correlation between ABCG2 and key immune cells; (C) bar chart analysis of the correlation between IL1RN and key immune cells; (D) bar chart analysis of the correlation between REG4 and key immune cells; (E) bar chart analysis of the correlation between SERPINB5 and key immune cells; (F) bar chart analysis of the correlation betweenTRIM29 and key immune cells.

Expression of the key genes in UC mouse model

Compared with mice in the NC group, C57BL/6 mice treated with 5% or 7% DSS showed a significant decrease in body weight (Fig. 8A), visible colon shortening and colonic damage (Fig. 8B), as well as significantly increased score of DAI (Fig. 8C). H&E staining results demonstrated significant inflammatory cell infiltration, loss of crypts, and epithelial destruction in the colonic epithelium (Fig. 8D). These results indicated the successful construction of the UC mouse model. The expression levels of the five key genes in the colon tissues were tested by qRT-PCR. Compared with the control group, the expression of ABCG2 was down-regulated, and the expression of IL1RN, REG4, SERPINB5 and TRIM29 were up-regulated in the model group (Fig. 8E). Furthermore, a clear DSS concentration-dependent trend was observed, revealing a more severe disease state of the UC mice treated with 7% DSS compared to those by 5% DSS, with more pronounced changes in gene expression.

Figure 8 Establishment of the animal model and detection of key gene expression.

(A) Changes in the body weight of mice in each group; (B) representative maps of the colon tissue in each group; (C) disease activity index (DAI) of mice in each group; (D) representative diagram of H & E staining of the colon tissue of mice in each group; (E) relative expression levels of the key genes (ABCG2, IL1RN, REG4, SERPINB5 and TRIM29) (qRT-PCR); (compared with the NC group, *P < 0.05, **P < 0.01, ***P < 0.001).

Impact of the key genes on cell viability and apoptosis in UC cell model

Relative to the control group, we found that ABCG2 overexpression significantly increased cell activity (Fig. 9A), and concurrently markedly increased the protein expression level of Bcl-2 but reduced the protein expression levels of Bax and Cleaved Caspase-3 (Fig. 9B). TUNEL staining further confirmed a notable reduction in apoptosis following ABCG2 overexpression (Fig. 9C). In addition, knockdown of IL1RN, REG4, SERPINB5 and TRIM29 caused a remarkable raise in cell activity and a substantial decrease in apoptosis in comparison to the control group (Figs. 9A–9C).

Figure 9 Impact of the identified key genes on cell function in the constructed UC cell model.

(A) Cell activity (%) detected by CCK-8; (B) the expression levels of apoptosis-related proteins (Bax, Bcl-2 and Cleaved Caspase-3) detected by Western blot. (C) Cell apoptosis in each group detected by TUNEL staining (compared with the control group, *P < 0.05, **P < 0.01, ***P < 0.001).

Discussion

UC is a form of inflammatory bowel disease marked by persistent and diffuse mucosal inflammation in the colon, extending from the proximal rectum (Feuerstein, Moss & Farraye, 2019). However, many existing therapeutic drugs for UC often disrupt metabolism and immune responses, frequently causing severe adverse reactions (Fan et al., 2021). Hence, the search for new therapeutic targets for UC is imperative. In this study, we conducted an analysis of the DEGs in GSE38713, GSE42911 and GSE134025 datasets, and ultimately screened out a total of 44 DEGs, encompassing 34 up- and 10 down-regulated ones.

Immune response is crucial in the onset, enhancement and continuation of UC (Bullard et al., 2022). The loss of immune tolerance leads to an increase in diverse pro-inflammatory mediators secreted by various types of cells, thus causing inflammation (Tatiya-Aphiradee, Chatuphonprasert & Jarukamjorn, 2018). Cytokines are critical in the inflammatory process, functioning as cell signaling molecules that promote inflammation and the pathogenesis of UC by activating inflammatory pathways and producting inflammatory mediators (Billmeier et al., 2016). With the conduction of GO and KEGG functional enrichment analyses of the DEGs, it was found that the function of these genes were primarily associated with pathways including immune response, immune response ligand activity, and signal receptor activator activity through GO analysis, and their major enrichment were also revealed in pathways such as cytokine–cytokine receptor interaction, the IL-17 signaling pathway, the TNF signaling pathway, and the chemokine signaling pathway by KEGG analysis.

According to the functional enrichment analysis results, we found that the functions of these DEGs were primarily related to immune responses, suggesting their potential role in regulating the progression of UC. Subsequently, through analysis of the key genes, ABCG2, IL1RN, REG4, SERPINB5 and TRIM29 were finally screened out. ABCG2 (transporter) is an export ABC protein that can expel a wide range of chemically unrelated heterologous and endogenous substances from cells (Gyongy et al., 2023). IL1RN(Interleukin-1 receptor antagonist gene) serves as a natural antagonist of IL-1. It encodes IL1 antagonist protein (IL1RA) and also plays an important role in the development of non-alcoholic fatty liver disease (Chen et al., 2022b). REG4 (regenerated islet derived 4 gene), identified through high-throughput sequencing of ulcerative colitis cDNA library, binds to and kills inflammatory Escherichia coli by activating epidermal growth factor receptor (EGFR) /Akt/cAMP response elements. This action enhances macrophage polarization toward M2, and is closely related to infection and inflammation, and tumorigenesis (Zheng, Xue & Zhang, 2022). SERPINB5 (serine protease inhibitor b5), initially identified in breast cancer, serves as a tumor suppressor by suppressing metastasis and angiogenesis, as well as promoting cell adhesion and apoptosis (Mahananda et al., 2021). TRIM29, encoded by the ATDC gene, plays crucial roles in diverse cellular processes, including transcriptional regulation, signal transduction, innate immunity and programmed cell death (Hsu, Yanagi & Ujiie, 2021). To further elucidate the link between these five key genes and the important immunocytes, immunoinfiltration analysis was performed, with results indicating their primary correlation with major immune-regulatory cells, including dendritic cell activation, mast cell activation, macrophage M1, neutrophils, macrophages M2, T-cell regulation (Tregs), and eosinophils. Therefore, it was concluded that they may regulate the occurrence and development of UC by regulating the body’s inflammatory responses.

The expression levels of the five key genes were assessed in GSE87466, revealing that ABCG2 was down-regulated, whereas IL1RN, REG4, SERPINB5 and TRIM29 were up-regulated in UC patients. To make further validation on the expression of these five key genes in UC, we constructed a UC mouse model. As is known, DSS is a chemical inflammatory agent that can induce inflammatory bowel disease in mice when it is added to the drinking water of them (Yan et al., 2018). Reportedly, a known UC mouse model construction was once completed by adding DSS to the drinking water of adult mice (Chen et al., 2022a), which closely simulated the clinical phenotype of human UC. Once the UC model is successfully created, symptoms like bloody stools, diarrhea, and weight loss can be observed (Nunes et al., 2019). Furthermore, this model can induce typical pathological characteristics of colitis, including mucosal obstacle damage and immunocyte infiltration (Zhao et al., 2020; Peng et al., 2020). Therefore, in this study, DSS induction was used for UC mouse model construction. By detecting the expression levels of ABCG2, IL1RN, REG4, SERPINB5 and TRIM29 in this constructed model, it was discovered that the model group exhibited down-regulation of ABCG2 and up-regulation of IL1RN, REG4, SERPINB5 and TRIM29 in comparison to the NC group, which was consistent with bioinformatics analysis results.

Researches have indicated a remarkable reduction in the level of ABCG2 expression during active inflammation, while TNF- α has been shown to significantly alter ABCG2 transcription in isolated differentiated epithelial organoids (Englund et al., 2007; Ferrer-Picon et al., 2020). IL1RN regulates the extent and duration of UC inflammation and also reduces joint swelling and damage in animal models of rheumatoid arthritis (Ashwood et al., 2004; Thompson, Dripps & Eisenberg, 1992). REG family proteins are activated in colitis, and REG4 has been proved as a target for intestinal epithelial stem cell enrichment and mucosal healing  (Granlund et al., 2011; Nishimura et al., 2019). Alterations in SERPINB5 expression have been observed in UC epithelial cells (Planell et al., 2013), while TRIM29 has been demonstrated in promoting pyroptosis of diabetic nephropathy podiocytes through the NF-kB/NLRP3 inflammasome pathway (Xu et al., 2023). These studies suggest that these key genes may be involved in UC progression by regulating inflammation and apoptosis-related pathways. In this study, a UC cell model was constructed by stimulating Caco-2 cells with LPS. Upon overexpression of ABCG2 or knockdown of IL1RN, REG4, SERPINB5 and TRIM29 at the cellular level, a significant increase in cell activity and a notable decrease in apoptosis levels were observed, indicating that these key genes may indeed play a regulatory role in UC development by regulating apoptosis-related pathways.

Conclusion

By screening UC-related key genes through bioinformatics analysis, followed by immune infiltration analysis, combined with validation in both UC mouse model and cell model, this study provides novel insights for finding new therapeutic targets for UC, holding significant scientific significance. However, it is also important to note that no investigation was carried out in this study into the specific effects and mechanisms of these key genes in UC, thus needing further in-depth research in this regard in the future.

Supplemental Information

Supplemental Information 1 Supplementary Figures

Supplemental Information 2 MIQE checklist

Supplemental Information 3 Raw numerical data for Figures 5 and 7

Supplemental Information 4 ARRIVE 2.0 Checklist

Additional Information and Declarations

Competing Interests

Author Contributions

Animal Ethics

Data Availability

The authors declare there are no competing interests.

Siyi Ni conceived and designed the experiments, analyzed the data, authored or reviewed drafts of the article, and approved the final draft.

Yingchao Liu performed the experiments, analyzed the data, prepared figures and/or tables, and approved the final draft.

Jihong Zhong performed the experiments, prepared figures and/or tables, and approved the final draft.

Yan Shen conceived and designed the experiments, authored or reviewed drafts of the article, and approved the final draft.

The following information was supplied relating to ethical approvals (i.e., approving body and any reference numbers):

The Animal Care and Use Committee (IACUC) of Zhejiang Laboratory Animal Committee (ZJCLA) (approval number: ZJCLA-IACUC-20020046).

The following information was supplied regarding data availability:

The data used and/or analyzed during the current study are available in this submission.

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
