# Peer review of "Identification and immunoinfiltration analysis of key genes in ulcerative colitis using WGCNA"

_PeerJ, doi:10.7717/peerj.16921_

## Round 0.1 · original submission · Major Revisions

Please address concerns of both reviewers and revise manuscript accordingly.

·

Basic reporting

1. The manuscript is badly written, and this makes it very difficult to read. English grammar and syntax have to be improved throughout the manuscript.
2. Intro & Background:
1) Unmet medical needs have not been clearly addressed.
a) In clinical setting, the diagnosis of UC is not difficulty. The clinical medical needs of the biomarker for UC seems not very meaningful.
b) Or the author try to emphasize the needs of biomarker for UC at EARLY stage? (line 52) If this is the case, suggest to introduce the clinical needs for EARLY diagnosis of UC, which could not be implemented by current tests? And could the EARLY diagnosis lead to better treatment outcome with less cost?
c) Literatures have not been well referenced, for example: Line 48-51, no reference be cited.
3. Figures descriptions are not clear enough. For example: A brief overall description of each figure should be provided, rather than just a wording, eg. Figure 1, “Screening of DEGs” of XXX for XXX?; Figure 2 “WGCNA” of what? Figure 3 “module gene analysis” of XXXX? et al..

Experimental design

no comments

Validity of the findings

1. The manuscript report a novel result. However, the clinical impact seems relative limited, considering the diagnosis of UC in clinical is not difficulty in clinical setting.
2. Data provided are fine.
3. The data could not support the conclusion “The results of this research hold clinical significance by offer novel insights for UC treatment” (line 287-288), as there is not intervention-related data have been investigated for these identified genes.

Additional comments

NA

Reviewer 2 ·

Basic reporting

The logic is clear, and English describe is suitable.

Experimental design

The experimental design is suitable.

Validity of the findings

The findings is novel

Additional comments

In this study, the authors mainly focus on that identification and immunoinfiltration analysis of key genes in ulcerative colitis using WGCNA method. Finally, the authors screened ABCG2, IL1RN, REG4, SERPINB5 and TRIM29 as key genes related to UC by bioinformatics analysis. Interestingly, the expression of ABCG2 was down-regulated when IL1RN, REG4, SERPINB5 and TRIM29 were up-regulated in UC mice. Overall, the logic is clear, and the data is solid. However, there are some issues that should be addressed, thus the manuscript could not be accepted for publication in PeerJ in its current form.

1: In Figure 8E, the authors should test the gene’s protein level.

2: There are no detail molecular mechanism in this manuscript, at least, the author should discuss the potential mechanisms in discussion.

3: It will be better to analyze the function of ABCG2, IL1RN, REG4, SERPINB5, and TRIM29 when knock down or overexpression in cell level, such as cell growth.

4: There are some spelling/grammatical errors in this manuscript, which should be corrected and modified.

---

## Round 0.2 · accepted · Accept

All issues pointed out by the reviewers were adequately addressed and the revised manuscript is acceptable now.

Reviewer 2 ·

Basic reporting

The author has answered and resolved almost all of the questions reviewers mentioned.

Experimental design

The design is reasonable.

Validity of the findings

Novel